# Re-ViLM: Retrieval-Augmented Visual Language Model for Zero and Few-Shot Image Captioning

**Zhuolin Yang*‡[1]** **Wei Ping*[2]** **Zihan Liu[2]** **Vijay Korthikanti[2]** **Weili Nie[2]**

**De-An Huang[2]** **Linxi Fan[2]** **Zhiding Yu[2]** **Shiyi Lan[2]** **Bo Li[1]** **Mohammad Shoeybi[2]**

**Ming-Yu Liu[2] Yuke Zhu[2,3] Bryan Catanzaro†[2] Chaowei Xiao†[2,4] Anima Anandkumar†[2,5]**

## Abstract

Augmenting pretrained language models (LMs) with a vision encoder (e.g., Flamingo) has obtained the state-of-the-art results in image-to-text generation. However, these models store all the knowledge within their parameters, thus often requiring enormous model parameters to model the abundant visual concepts and very rich textual descriptions. Additionally, they are inefficient in incorporating new data, requiring a computational-expensive fine-tuning process. In this work, we introduce a **Re**trieval-augmented **Vi**sual **L**anguage **M**odel, **Re-ViLM**, built upon the Flamingo, that supports retrieving the relevant knowledge from the external database for zero and in-context few-shot image-to-text generations. By storing certain knowledge explicitly in the external database, our approach reduces the number of model parameters and can easily accommodate new data during evaluation by simply updating the database. We also construct an interleaved image and text data that facilitates in-context few-shot learning capabilities. We demonstrate that Re-ViLM significantly boosts performance for image-to-text generation tasks, especially for zero-shot and few-shot generation in out-of-domain settings with $4\times$ less parameters compared with baseline methods.

## 1 Introduction

Image-to-text generation, also known as image captioning (e.g., Karpathy and Fei-Fei, 2015), plays a vital role in understanding visual information and enhancing human-AI interaction. This task has a wide range of practical applications such as gaming, virtual reality, and robotics (Luo et al., 2019; Zhao et al., 2021; Liu et al., 2021). To address this

---

*Equal contribution. ‡Work done during an internship at NVIDIA. [1]UIUC. [2]NVIDIA. [3]UT Austin. [4]University of Wisconsin–Madison. [5]California Institute of Technology. †Equal advising. Correspondence to: Zhuolin Yang <zhuolin5@illinois.edu>, Wei Ping <wping@nvidia.com>

problem, numerous methods have been proposed recently and obtained great success (e.g., Alayrac et al., 2022; Wang et al., 2022c; Hu et al., 2022; Chen et al., 2022b,a; Wang et al., 2022b).

Among them, visual language models (LMs) build on top of pretrained autoregressive LMs (e.g., GPT-3, Brown et al., 2020), and inherit its powerful text generation ability. In particular, the pretrained LM parameters are usually frozen and only some trainable layers (e.g., adaptor) are added into the large LM during multimodal pretraining (Eichenberg et al., 2022; Mokady et al., 2021; Tsimpoukelli et al., 2021; Alayrac et al., 2022). This frozen LM strategy can avoid catastrophic forgetting when the visual LM is trained on ⟨*image, text*⟩ data, where the text quality is usually lower than the text-only corpus to pretrain LM. In addition, it enables the compelling zero-shot or few-shot capability of pretrained LM (e.g., Flamingo, Alayrac et al., 2022).

However, such methods have two major limitations: 1) They store all acquired knowledge within the model parameters, making them parameter inefficient in modeling the abundant visual concepts (e.g., uncommon objects) and rich textual descriptions (e.g., alternative descriptions for the same scene). 2) They are inefficient in incorporating new data, typically requiring computationally expensive fine-tuning (Chen et al., 2022a) or pertaining on increasingly more parameters and interleaved ⟨*image, text*⟩ data (Alayrac et al., 2022).

In the past few years, retrieval-augmented LMs (Guu et al., 2020; Lewis et al., 2020; Karpukhin et al., 2020; Borgeaud et al., 2022) have shown notable success in improving accuracy while reducing model parameters by retrieving large-scale text corpus. Despite their success, there are several issues to be addressed before we apply retrieval technique to visual LMs for image-to-text generation: *i)* The visual LM needs to seamlessly retrieve and encode external knowledge at the beginning of multi-modal pretraining. Otherwise, the

| Query Samples | Generated Captions | Retrieved Evidence | |
|---|---|---|---|
| 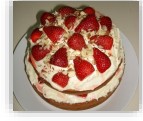 | **Flamingo:**
A brown and black animal in a pool of water.

**Re-ViLM:**
A sea otter sitting on the beach. | 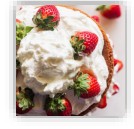
Sea otter (Enhydra lutris) floating in water holding a starfish. | A sea otter and its young swimming together on their backs. |

Figure 1: Examples of input images and output captions from 2.1B Flamingo (re-implemented) and 2.4B Re-ViLM. Re-ViLM can utilize the retrieved captions to generate more informative and accurate captions.

powerful pretrained autoregressive LM tends to ignore the poorly encoded external knowledge. *ii)* In multi-modal datasets, there are cases where multiple captions describing the same image (e.g., from different annotators (Lin et al., 2014)), and multiple images having the same caption (e.g., see images in Figure 3 in Appendix). Thus, simply performing standard nearest neighbor retrieval tends to make the model take a shortcut and copy-paste retrieval examples during training. *iii)* Training the model on large-scale interleaved ⟨*image, text*⟩ dataset facilitates few-shot learning capability (e.g., M3W in Alayrac et al. (2022)), but it is really expensive to collect such dataset.

In this work, we propose a **Re**trieval-augmented **Vi**sual **L**anguage **M**odel, **Re-ViLM**, which enhances the state-of-the-art visual LM, Flamingo for zero-shot and in-context few-shot image captioning,[1] by seamlessly incorporating a multimodal retriever and retrieval-augmented LM layers that cross-attend to a text encoder (see selected samples in Figure 1, model framework in Figure 2). Specifically, we make the following contributions:

1. In contrast to previous work, we initialize Re-ViLM with RETRO, a pretrained retrieval-augmented LM (Borgeaud et al., 2022), thus it can seamlessly integrate the retrieval capability

at the beginning of multimodal pretraining and result in improved performance.

2. We investigate the retrieval strategy to build the multimodal retriever. At multimodal pretraining, we find the best performance is obtained by retrieving *k*-nearest neighbor captions based on *cosine* similarity between image CLIP (Radford et al., 2021) embeddings, while circumventing "copy-and-paste" behavior in training by filtering out retrieved candidates with the same caption as the training instance.

3. We construct both pretraining and evaluation datasets consisting of interleaved ⟨*image, text*⟩ pairs for multimodal pretraining, using existing public datasets. This facilitates in-context learning where few-shot examples are given as interleaved ⟨*image, text*⟩ pairs.

4. We conduct extensive experiments for image-to-text generation under zero-shot, few-shot, and fine-tuning settings on various benchmarks including MSCOCO (Lin et al., 2014), Flickr30k (Plummer et al., 2015), and No-Caps (Agrawal et al., 2019). Re-ViLM consistently outperforms the baseline Flamingo model across all settings. The improvements are particularly notable in zero-shot and few-shot settings, e.g., our Re-ViLM can outperform the Flamingo model containing even $4\times$ more parameters in zero-shot evaluation.

We organize the rest of the paper as follows. In § 2,

---

[1]Note that, no official implementation of *Flamingo* is available, which is trained on large-scale in-house dataset (Alayrac et al., 2022). We re-implement the model on public available dataset.

we discuss related work. We introduce Re-ViLM model in § 3 and multimodal dataset for pretraining and retrieval in § 4. We present our experimental results in § 5 and conclude the paper in § 6.

## 2  Related Work

**Visual Language Models**  Many recent work tackle the problem of generating text captions for given images (e.g., Wang et al., 2022c; Alayrac et al., 2022; Aghajanyan et al., 2022; Wang et al., 2022b; Hu et al., 2022; Li et al., 2022; Chen et al., 2022b; Yu et al., 2022). Among these work, visual language models (Tsimpoukelli et al., 2021; Alayrac et al., 2022) directly augment pretrained LMs with visual component, achieving strong results in both zero-shot and few-shot generation.

**Retrieval-augmented Models** Retrieval has been successfully applied in various NLP tasks, including question answering (Guu et al., 2020; Karpukhin et al., 2020), autoregressive language modeling (Borgeaud et al., 2022), and other knowledge-intensive tasks (e.g., Lewis et al., 2020). In computer vision, retrieval has also been applied for image recognition with long-tail distribution of classes (Long et al., 2022). In this work, we apply retrieval for image captioning.

RA-CM3 (Yasunaga et al., 2022) augments the CM3 backbone (Aghajanyan et al., 2022) with retrieval, which can perform both image-to-text generation and text-to-image synthesis. In contrast to our ReViLM, there are the following differences: 1) We investigate the "copy-and-paste" behavior of retrieval-augmented model during training, and propose a simple filtering strategy during retrieval. In contrast, RA-CM3 proposes a query-dropout strategy that drops some tokens of the query caption used in retrieval. In our ablation study, we find our simple strategy works better than the dropout regularization as shown in § 5.5.1. 2) Our Re-ViLM uses retrieval-augmented LM decoder layer with cross-attention module to attend to the retrieved similar captions. In contrast, RA-CM3 appends the retrieved captions as the prefix context on the decoder side. In our ablation study, we find ours are more effective, as shown in § 5.5.2. 3) For image-to-text generation, RA-CM3 only provides the result of 2-shot in-context image captioning on MSCOCO (Lin et al., 2014). In contrast, we perform extensive evaluations of our Re-ViLM on various benchmarks under zero-shot, few-shot and fine-tuning settings.

## 3  Re-ViLM Architecture

In this section, we begin by outlining the framework of Re-ViLM. After that, we delve into the details of each component in depth.

### 3.1  Framework

We illustrate our Re-ViLM framework for image captioning in Figure 2. It consists of three essential components:

- *Image encoder* begins with a pretrained vision transformer from CLIP (Radford et al., 2021) to extract visual features from the input images. These features are subsequently fed into a trainable perceiver resampler (Jaegle et al., 2021) to unify the image features into the textual representation used in the retrieval-augmented LM.

- *Retrieval-augmented LM* is initialized with a pretrained RETRO model (Borgeaud et al., 2022) to generate corresponding captions based on the image features and retrieved evidence. Among some LM layers, the module cross-attend the hidden representations from the image encoder or the shared text encoder. The bidirectional text encoder encodes the retrieved captions obtained from the multimodal retriever.

- *Multimodal retriever* consists of a retrieval database storing ⟨*image, text*⟩ pairs indexed by Faiss, a fast similarity search library (Johnson et al., 2019) using their CLIP embeddings. Given a query image, the retriever extracts its embedding using the CLIP-ViT module within the image encoder. It then returns the top-*k* ⟨*image, text*⟩ pairs, measured by the cosine similarity of embeddings between the query image and the retrieved images. After that, the retrieved captions are encoded by the LM to generate relevant captions.

In the following subsections, we provide more details about each component of our model.

### 3.2  Image Encoder

To fully leverage the existing pretrained model, we initialize the image encoder with CLIP-ViT (Radford et al., 2021), a pretrained vision transformer that processes images by dividing them into a grid of patches and then processing each patch with a transformer encoder. In our experiments, we used two different sizes of the CLIP-ViT model: ViT-B/32 and ViT-L/14. We freeze the CLIP-ViT part during multimodal pretraining to avoid catastrophic forgetting while making it trainable during

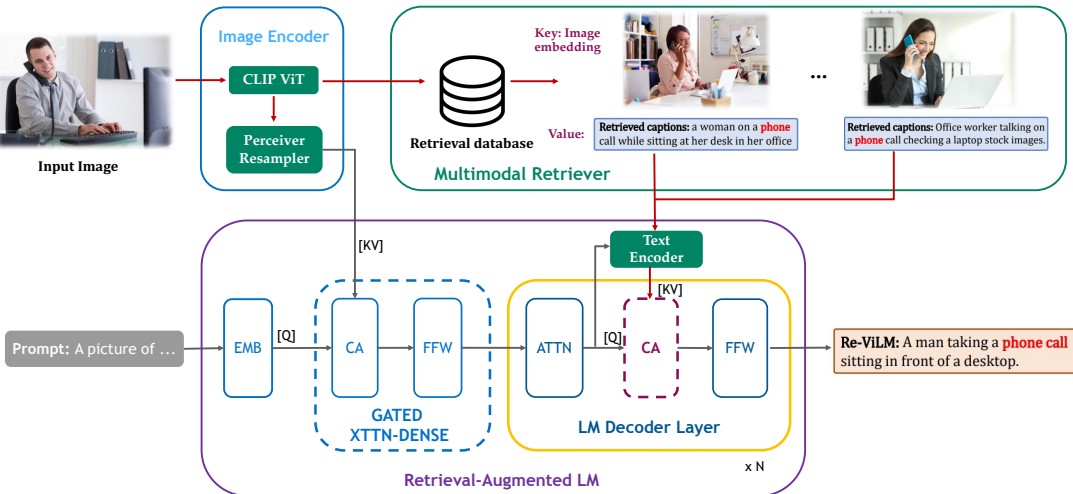

Figure 2: The framework of Re-ViLM. The model first extracts CLIP embedding of the input image, and use it to retrieve similar image-text pairs from the database. Within some predetermined layers, the *retrieval-augmented LM* will cross-attend the visual representation from the image encoder, and the textual representation from text encoder, which encodes the retrieved captions.

fine-tuning for better results. We then use a perceiver resampler to obtain fixed-length hidden representations from the CLIP-ViT token embeddings. The perceiver is trainable during both the multimodal pretraining and future fine-tuning, allowing it to adapt the visual representations for the text decoder and connects the two modalities.

### 3.3 Multimodal Retriever

**Building Database.** The retrieval database is built upon a image-text paired dataset and is structured as a key-value map, where the keys are CLIP ViT-B/32 image embeddings, and the values are the corresponding text descriptions. The database is indexed by Faiss library (Johnson et al., 2019).

**Retrieval** Given an input query image $I$, we perform $k$-nearest neighbor retrieval with cosine similarity of embeddings between query image and database images.[2] The retrieval results are denoted as $\mathcal{R}(I) = \{(i_1, c_1), \cdots, (i_k, c_k)\}$, where $i_j$ and $c_j$ with $j \in [1, k]$ represents the retrieved image and caption, respectively.

**Filtering strategy** There can be multiple captions for the same image from different annotators in some datasets (e.g., MSCOCO (Lin et al., 2014)). In this case, these retrieved captions, which are highly correlated (even near-duplicate) with the ground-truth caption, could give a false sense of retrieved evidence quality to the model, resulting

in potentially degraded test performance due to the discrepancy between the training and evaluation setting. To avoid it, we filter out the retrieved image-text pairs if the retrieved image is identical to the query image $I$ (e.g., $i_1 = I$) during both training and inference.

Furthermore, in the image-text datasets (e.g., Conceptual Captions (Sharma et al., 2018; Changpinyo et al., 2021)), multiple images can have identical captions. For example, one annotator may provide the same caption or alt-text for similar images (see Figure 3 in Appendix for examples). If the retrieved caption from database is the same as the ground-truth caption during training, it will encourage the model to take a short path and simply copy-and-paste the retrieved caption to the model output, hindering the training of Re-ViLM. To address this issue, we employ a filtering strategy that filters out the retrieved image-text pair if its text is identical to the training image's caption that is used as the teacher-forced input at the LM decoder layer. Thus, if the corresponding ground-truth caption of $I$ is $C$, the filtered retrieval results $\mathcal{R}(I) = \{(i_1, c_1), \cdots, (i_k, c_k) \mid c_j \neq C, i_j \neq I, \forall j \in [1, k]\}$. In § 5.5.1, we show that Re-ViLM can be largely improved with this simple filtering strategy.

### 3.4 Retrieval-augmented LM

To facilitate the model using the retrieved captions, the visual LM needs to seamlessly retrieve and encode the external knowledge at the beginning of

---

[2] We also tried to use embeddings of query image and database captions with the same CLIP model, and the empirical results are very similar.

multimodal pretraining. Thus, we initialize our text encoder and LM decoder layer with pretrained RETRO (Borgeaud et al., 2022), a state-of-the-art retrieval-augmented LM. In our experiments, we use three different sizes of RETRO models: RETRO$_{base}$ with 148M parameters, RETRO$_{medium}$ with 405M, and RETRO$_{large}$ with 1.5B parameters. To generate captions conditioned on visual input, we interleave the LM decoder layers with gated cross-attention dense layers (gated xttn-dense) as in *Flamingo* (Alayrac et al., 2022), which take the output of perceiver resampler as the *key* and *value* for cross-attention. To incorporate the retrieved captions as evidence, we interleave the LM decoder layers with retrieval-augmentation layers, which take text encoder output as *key* and *value* for cross-attention. Note that, the text encoder is shared across retrieval-augmentation layers. We freeze the retrieval-augmented LM at multimodal pretraining, and make it trainable at fine-tuning.

**Text encoder** Given the retrieved $k$-nearest neighbor captions, we use a transformer-based bidirectional encoder model, to obtain the hidden representations. Specifically, our text encoder shares the subword embedding table with the LM decoder. We concatenate $k$ embeddings along with the length dimension to form the retrieval raw embedding tensor $\mathcal{E} \in \mathbb{R}^{(k \times m) \times d}$, where $m$ is sequence length, $d$ is hidden dimension. After applying transformer layers, the encoder output is cross-attended by the LM decoder layers. We initialize the text encoder with a pretrained RETRO model (Borgeaud et al., 2022) instead of training from scratch to improve model performance.

### 3.5 Trainable Modules at Pretraining and Finetuning

Similar to Flamingo, our training objective is to maximize the conditional likelihood of the captions given the images. At pretraining, we follow Flamingo's strategy by freezing the pretrained components, including CLIP-ViT and retrieval-augmented LM, training only the perceiver resampler from scratch. At finetuning, we unfreeze all the pretrained components and increase image resolution from $224 \times 224$ to $480 \times 480$, as suggested in (Alayrac et al., 2022). This has been shown to improve overall performance.

## 4 Multimodal Data for Pretraining and Retrieval

### 4.1 Image-Text Pair Data

In this work, we pretrain our models using two multi-modal datasets: 1) **CC3M + CC12M + SBU**, which consists of overall 15 million high-quality image-text pairs from the Conceptual Dataset (Sharma et al., 2018; Changpinyo et al., 2021) and SBU Captions (Ordonez et al., 2011); 2) **COYO-700M** (Byeon et al., 2022), which contains 747 million image-text pairs after filtering out low-quality samples from a collection of 10 billion web image-text sources. In our experiment, we find that the high-quality captions are essential for pretraining in image captioning task. Thus, we further filter out instances with irregular textual tokens or low CLIP similarity scores between image and text, and obtain 104M high-quality image-text pairs. For simplicity, we refer to the CC3M + CC12M + SBU dataset as **CCS** and the COYO-104M dataset as **COYO**. At retrieval, we use **CCS** and **COYO** as the main sources for our retrieval database, and utilize the Faiss library (Johnson et al., 2019) to support fast similarity-based retrieval. It takes 241GB for storing faiss index file and for each query, it takes around 50ms performing retrieval on our database.

### 4.2 Interleaved Image-Text Data

In this subsection, we discuss our proposed pretraining method for enhancing the in-context few-shot ability of our model. In this scenario, the model needs to be highly conditioned on the previous few-shot samples (image-text pairs) to effectively generate captions of test images. However, existing multimodal models generally do not use multiple image-text pairs as inputs for pretraining (Tsimpoukelli et al., 2021; Yasunaga et al., 2022). This makes the in-context few-shot learning at inference time challenging, as there is no such supervision during pretraining. While Alayrac et al. (2022) built an in-house large-scale multimodal corpora with interleaved images and text, collecting such a dataset is expensive.

We construct our image-text interleaved datasets using publicly available image-text pair datasets. We make the image-text pairs in each interleaved sample relevant, in order to explicitly teach the model how to condition on previous data samples for generating the caption of the current image.

Our interleaved dataset is constructed by using **CCS**. For each image-text pair (query) in **CCS**, we

select four relevant data pairs from the same corpus to construct each interleaved sample, which results in five data pairs for each interleaved sample. The data selection process for each query consists of two steps. **Step-1**: We use $L_2$ metric to measure the distances between the CLIP embeddings of the query image and the rest of images in CCS, and select data pairs where the images have a normalized distance score between $0.4$ and $0.6$ to the query image.[3] **Step-2**: To ensure the captions in an interleaved sample are similar, we further use CLIP embeddings to calculate the distances between the query caption and the captions from the selected data pairs. We pick the top-4 data pairs where the captions are the most similar to the query caption.

## 5 Experiments

In this section, we evaluate the performance of Re-ViLM under three different settings: zero-shot, few-shot and fine-tuning, on various image captioning benchmarks. We aim to demonstrate the superiority of our retrieval augmentation technique in improving the quality and relevance of generated captions through retrieving relevant knowledge from external databases. We compare our results to several widely-used image captioning models such as Enc-Dec (Changpinyo et al., 2021), SimVLM (Wang et al., 2022c), and Flamingo (Alayrac et al., 2022). Through extensive evaluation of Re-ViLM, we conclude that Re-ViLM is compelling under zero-shot and few-shot settings.

### 5.1 Experimental setup

**Evaluation Dataset.** We conduct our image captioning evaluation on three multi-modal datasets: **1) MSCOCO** (Lin et al., 2014) is a dataset for image captioning, object detection, and segmentation. We use the Karpathy split (Karpathy and Fei-Fei, 2015), with 82k/5k/5k images for training, validation, and testing respectively. Each image is annotated with at most 5 human-generated captions. **2) Flickr30k** (Plummer et al., 2015) is a standard benchmark for sentence-based image captioning, which includes 29k/1k/1k images in its Karpathy split. **3) NoCaps** contains 15k images containing nearly 400 additional novel classes to

original MSCOCO, which can be used to evaluate novel object captioning performance after finetuning on MSCOCO. For zero-shot setting, we focus on the evaluation under MSCOCO and Flickr30k datasets. For fine-tuning setting, we evaluate on MSCOCO, Flickr30k and NoCaps datasets. We conduct our few-shot experiments on MSCOCO dataset only to assess Re-ViLM's generalization and adaptability. Throughout our experiments, we report BLEU@4, CIDEr, and SPICE scores (Lin et al., 2014) to measure the quality and relevance of the generated captions given input images.

**Implementation.** We develop Re-ViLM with different scales based on different size of CLIP-ViT and RETRO, named Re-ViLM$_{base}$ (ViT-B/32, RETRO-148M), Re-ViLM$_{medium}$(ViT-L/14, RETRO-410M) and Re-ViLM$_{large}$(ViT-B/32, RETRO-1.5B). Compared to Flamingo model with the same CLIP-ViT and comparable GPT-3 configuration (Brown et al., 2020), Re-ViLM introduces up to $16\%$ additional parameters while largely boosting the performance. We build our model with Megatron-LM infrastructure to support large visual LM training and evaluation. We set global batch size as 256 and use Adam optimizer at training. We use beam search with beam size as 3, maximum generation length as 10 for inference. In our experiments, we set the number of retrieved captions $k = 2$ for Re-ViLM. We also evaluate Re-ViLM performance with larger $k = 5, 10$, which indicates insignificant improvements. Details can be found in Appendix B.

### 5.2 Zero-shot Evaluation

We conduct zero-shot evaluation on MSCOCO and Flickr30k datasets. During pretraining, we include both CCS and COYO as our retrieval database and report the best number among all different settings. Results are shown in Table 1. We find that, Re-ViLM could achieve significant boosts (around 10.0 on CIDEr score) compared to the Flamingo model, by introducing up to $16\%$ additional parameters. Even Re-ViLM$_{base}$ outperforms the largest SimVLM by a large margin. We leave the full results containing Re-ViLM's performance under different pretraining and retrieval database combination in Appendix A.

### 5.3 Few-shot Evaluation

We evaluate the few-shot learning capability of Re-ViLM by pretraining it on the constructed interleaved CCS dataset, and evaluating it under the

---

[3]Note that we eliminate the most similar images since there are lots of near-duplicate images, e.g., they have very few differences in terms of resizing, cropping, color, rotation, watermark. In practice, we set thresholds $[0.4, 0.6]$ to filter out these cases, increase the diversity of the data pairs in one interleaved sample, and still make sure the selected images are relevant.

Table 1: Zero-shot evaluation results on MSCOCO, Flickr30k benchmarks, compared with different image captioning baselines. We report BLEU@4, CIDer, SPICE scores for different methods. Note that MSCOCO, Flickr30k were excluded from pretraining set in the following MSCOCO and Flickr30k results. We replicate Flamingo models with the same image encoder and text decoder as Re-ViLM based on original paper.

| Method | Total params. | Trainable params. | MSCOCO karpathy test | | Flickr30k karpathy test | |
|---|---|---|---|---|---|---|
| | | | BLEU@4 | CIDer | CIDer | SPICE |
| VL-T5 (Cho et al., 2021) | 224M | 224M | - | 4.9 | 2.6 | 2.0 |
| Unfied VLP (Zhou et al., 2020) | 122M | 122M | - | - | 24.9 | 7.2 |
| SimVLM$_{base}$ (Wang et al., 2022c) | - | - | 9.5 | 24.0 | - | - |
| SimVLM$_{large}$ | - | - | 10.5 | 24.9 | - | - |
| SimVLM$_{huge}$ | $\sim 1.4$B | $\sim$1.4B | 11.2 | 32.2 | - | - |
| Flamingo$_{base}$(re-impl) | 364M | 102M | 12.4 | 39.6 | 42.2 | 7.9 |
| Flamingo$_{medium}$(re-impl) | 894M | 233M | 15.6 | 44.3 | 43.2 | 8.8 |
| Flamingo$_{large}$(re-impl) | 2.1B | 489M | 16.5 | 49.2 | 46.4 | 9.4 |
| Re-ViLM$_{base}$ | 420M | 158M | 17.0 | 51.2 | 45.2 | 9.2 |
| Re-ViLM$_{medium}$ | 1.0B | 347M | 17.9 | 53.6 | 52.0 | 9.8 |
| Re-ViLM$_{large}$ | 2.4B | 806M | 18.6 | 60.8 | 52.1 | 10.0 |

Table 2: Few-shot evaluation results on MSCOCO benchmarks, compared with vanilla Flamingo models as our baseline. We report BLEU@4, CIDer scores for different methods. We pretrain our Re-ViLM on constructed **CCS** interleaved dataset and evaluate on constructed COCO interleaved dataset respectively. We adopt **CCS** as our retrieval set during both pretraining and evaluate stage.

| Method | Total params. | Trainable params. | 2 shots | | 4 shots | | 8 shots | |
|---|---|---|---|---|---|---|---|---|
| | | | BLEU@4 | CIDer | BLEU@4 | CIDer | BLEU@4 | CIDer |
| Flamingo-3B (Alayrac et al., 2022) | 3.2B | 1.3B | - | - | - | 85.0 | - | - |
| Flamingo-9B | 9.3B | 1.6B | - | - | - | 93.1 | - | - |
| Flamingo$_{base}$(re-impl) | 364M | 102M | 13.7 | 53.9 | 19.5 | 66.0 | 22.1 | 71.8 |
| Re-ViLM$_{base}$ | 420M | 158M | 14.8 | 60.1 | 20.8 | 72.2 | 21.8 | 72.6 |
| Flamingo$_{medium}$(re-impl) | 894M | 233M | 17.9 | 69.0 | 23.3 | 80.2 | 23.1 | 76.8 |
| Re-ViLM$_{medium}$ | 1.0B | 347M | 18.2 | 73.6 | 24.0 | 84.5 | 24.1 | 81.0 |
| Flamingo$_{large}$(re-impl) | 2.1B | 489M | 18.2 | 71.6 | 25.7 | 89.2 | 26.3 | 89.1 |
| Re-ViLM$_{large}$ | 2.4B | 806M | 18.4 | 77.2 | 25.5 | 90.5 | 26.2 | 90.2 |

Table 3: Finetuning evaluation results on MSCOCO, Flickr30k, and NoCaps benchmarks, compared with different image captioning baselines. Note that, for NoCaps, we finetune on MSCOCO karpathy train, following prior works (Li et al., 2022), while some work mentioning this setting as zero-shot evaluation. We finetune our Re-ViLM on MSCOCO/Flickr30k karpathy train split respectively for MSCOCO and Flick30k evaluation. We report BLEU@4, CIDer, SPICE scores for different methods.

| Method | Total params. | MSCOCO karpathy test | | Flickr30k karpathy test | | NoCaps validation | |
|---|---|---|---|---|---|---|---|
| | | BLEU@4 | CIDer | BLEU@4 | SPICE | CIDer | SPICE |
| Enc-Dec (Changpinyo et al., 2021) | - | - | 110.9 | - | - | 90.2 | 12.1 |
| VinVL (Zhang et al., 2021) | - | 38.2 | 129.3 | - | - | 92.5 | 13.1 |
| VL-T5 (Cho et al., 2021) | 172M | 34.6 | 116.1 | - | - | 4.4 | 5.3 |
| MetaLM (Hao et al., 2022) | 545M | 37.6 | 126.6 | - | - | 58.7 | 8.6 |
| Unfied VLP (Zhou et al., 2020) | 122M | 36.5 | 116.9 | 30.1 | 17.0 | - | - |
| BUTD (Anderson et al., 2018) | - | 36.2 | 113.5 | 27.3 | 16.0 | - | - |
| NBT (Lu et al., 2018) | - | 34.7 | 107.2 | 27.1 | 15.6 | - | - |
| SimVLM$_{huge}$ (Wang et al., 2022c) | $\sim$1.4B | 40.6 | 143.3 | - | - | 110.3 | 14.5 |
| BLIP (Li et al., 2022) | 252M | 38.6 | 129.7 | - | - | 105.1 | 14.4 |
| BLIP$_{CapFilt-L}$ (Li et al., 2022) | 252M | 40.4 | 136.7 | - | - | 113.2 | 14.8 |
| Flamingo$_{base}$(re-impl) | 364M | 37.0 | 128.0 | 30.4 | 16.5 | 102.8 | 14.0 |
| Flamingo$_{medium}$(re-impl) | 894M | 37.4 | 129.0 | 30.7 | 17.2 | 105.6 | 14.4 |
| Flamingo$_{large}$(re-impl) | 2.1B | 38.2 | 129.4 | 31.2 | 17.4 | 109.2 | 14.5 |
| Re-ViLM$_{base}$ | 420M | 37.8 | 129.1 | 30.6 | 17.3 | 105.2 | 14.2 |
| Re-ViLM$_{medium}$ | 1.0B | 38.2 | 131.2 | 31.0 | 17.5 | 106.8 | 14.4 |
| Re-ViLM$_{large}$ | 2.4B | 39.4 | 134.2 | 31.6 | 18.0 | 109.5 | 14.7 |

interleaved MSCOCO dataset, constructed by the same process described in § 4.2, with $\{2, 4, 8\}$-shots. Results are shown in Table 2. While the significant improvements on $\{2, 4\}$-shots setting compared with the comparable size Flamingo model are clearly observed, we notice that the retrieval augmentation benefits becomes less when the number of shots increases (i.e., 8-shot). This is not surprising as the few-shot in-domain examples from MSCOCO has more useful information to boost the model performance on MSCOCO, than the out-of-domain samples from our retrieval database, **CCS** and **COYO**. As the number of in-domain examples increases, the benefit of retrieval from out-of-domain examples becomes marginal.

## 5.4   Fine-tuning Evaluation

We conduct fine-tuning evaluation of Re-ViLM on MSCOCO, Flickr30K and NoCaps benchmarks. For evaluation on MSCOCO and Flickr30k, we fine-tune our pretrained Re-ViLM with smaller learning rate and early-stop strategy on MSCOCO and Flickr30k dataset respectively. For NoCaps evaluation, we fine-tune our model on MSCOCO dataset, following prior works (Li et al., 2022). Results are shown in Table 3. We observe that Re-ViLM still consistently outperforms Flamingo, although the relative improvements becomes smaller compared to the zero-shot and few-shot settings. We leave the full results containing Re-ViLM's performance under different pretraining and retrieval database combination in Appendix A.

## 5.5   Ablation Study

### 5.5.1   Filtering during Retrieval

There could exist two different types of duplication scenarios in multi-modal datasets: **Same image with multiple captions**, which is commonly found in MSCOCO, Flickr30k and NoCaps datasets, could lead to label leakage during training. **Multiple images with identical caption**, which is common in multimodal datasets such as Conceptual Captions, as shown in Figure 3 in Appendix C. [4] Both of these duplication can lead to a severe issue that Re-ViLM can simply copy and paste the retrieved captions to achieve $100\%$ match to the ground-truth captions. See Appendix C for more in-depth discussion.

To mitigate these above issues, we develop a

---

[4]We find that the ratio of identical captions in Conceptual Captions can be as high as $15.7\%$.

simple filtering strategy that discards retrieved samples that matches the training query image $I$ or its caption $C$ at training (i.e. $\mathcal{R}(I) = \{(i_1, c_1), \cdots, (i_k, c_k) \mid c_j \neq C, i_j \neq I, \forall j \in [1, k]\}$). We notice that another concurrent work RA-CM3, has also proposed the *query-dropout* strategy which mitigates such duplication issue by randomly dropping out retrieved caption tokens based on their similarity to the query image and text. We conduct ablation study to compare our simple filtering method with the query-dropout method. The results, as shown in Table 4, indicates that our simple filtering strategy leads to consistent improvement in the performance of Re-ViLM, while the query-dropout strategy achieves competitive but slightly worse results than simple filtering strategy.

Table 4: Comparison between ReViLM with simple filtering strategy, query-dropout strategy, and without any filtering method during retrieval. Models are pretrained on **CCS** dataset, and evaluated on MSCOCO, Flickr30k under zero-shot setting. We report B@4: BLEU@4, C: CIDer, S: SPICE scores for different methods.

| Method | MSCOCO | | Flickr30k | |
|---|---|---|---|---|
| | B@4 | C | C | S |
| Re-ViLM$_{base}$ [No filtering] | 12.3 | 35.5 | 41.4 | 8.1 |
| Re-ViLM$_{base}$ [Query-dropout] | 16.5 | 48.6 | 43.4 | 9.1 |
| Re-ViLM$_{base}$ | **17.0** | **51.2** | **45.2** | **9.2** |
| Re-ViLM$_{medium}$ [No filtering] | 12.3 | 35.5 | 41.4 | 8.1 |
| Re-ViLM$_{medium}$ [Query-dropout] | 17.5 | 52.1 | 50.5 | 9.6 |
| Re-ViLM$_{medium}$ | **17.9** | **53.6** | **52.0** | **9.8** |

### 5.5.2   Retrieval Augmentation as In-Context Prepending

Our Re-ViLM incorporate retrieved captions through the cross attention between the bidirectional text encoder and LM decoder layers. A concurrent work, RA-CM3, proposed an alternative retrieval augmentation method by appending the retrieved captions as the prefix context on decoder side as a simpler way to utilize retrieved evidence without introducing additional parameters. We investigate this prompt-like augmentation method, and replicate it by appending the top 2 retrieved evidence as prefix during pretraining and inference of Flamingo model. We compare this retrieval augmentation method with our retrieval-augmented LM layer approach. The results are shown in in Table 5. We can observe that our retrieval-augmented design has better zero-shot captioning performance than the retrieval augmentation method in Yasunaga et al. (2022). It reveals the importance of our retrieval-based architecture design.

Table 5: Comparison between different retrieval augmentation methods: retrieval-augmented LM layers (Re-ViLM) and in-context prepending as prompt (Flamingo + prepend), along with vanilla Flamingo model. Models are pretrained on CCS dataset, and evaluated on MSCOCO, Flickr30k under zero-shot setting. We report B@4: BLEU@4, C: CIDer, S: SPICE scores for different methods.

| Method | MSCOCO | | Flickr30k | |
|---|---|---|---|---|
| | B@4 | C | C | S |
| Flamingo$_{base}$ | 12.4 | 39.6 | 42.2 | 7.9 |
| Flamingo$_{base}$ + prepend | 13.4 | 43.4 | 43.5 | 8.2 |
| Re-ViLM$_{base}$ | **17.0** | **51.2** | **45.2** | **9.2** |
| Flamingo$_{medium}$ | 15.6 | 44.3 | 43.2 | 8.8 |
| Flamingo$_{medium}$ + prepend | 16.4 | 45.6 | 46.6 | 9.2 |
| Re-ViLM$_{medium}$ | **17.9** | **53.6** | **52.0** | **9.8** |

## 6 Conclusion

In this work, we propose Re-ViLM, a retrieval-augmented image-to-text model, with strong zero-shot and few-shot image captioning results. Re-ViLM, which is built on Flamingo, provides substantial reduction in the number of parameters while obtaining compelling results across different settings, as it does not need to store all knowledge within the parameters. We also propose a simple yet effective filtering strategy at retrieval to circumvent the "copy-and-paste" behavior of retrieval-augmented model. Furthermore, we construct an interleaved image-text dataset for pretraining, which is crucial for in-context few-shot learning. Extensive experiments on diverse image-captioning datasets shows that Re-ViLM consistently outperform the baseline Flamingo model across all settings. Additionally, we conduct experiments on fine-tuning settings and show promising results.

## 7 Limitations

In this paper, we focus on exploring emergent zero-shot and in-context few-shot image captioning. To achieve this, we designed our retrieval augmented model mainly based on the Flamingo framework (Alayrac et al., 2022), and leave the application of our retrieval design to other image-to-text frameworks (Bao et al., 2021; Chen et al., 2022b; Wang et al., 2022a) as future work. Furthermore, since there is no official implementation of Flamingo and its training datasets, our framework is based on our reimplemented Flamingo, trained on publicly available datasets and manually crafted interleaved image-text datasets. Also from the scaling perspective, comparable to GPT in the text-only domain, one of the most important advantages of the Flamingo-like model is scaling. In this study, we haven't been able to further scale Re-ViLM to 80B to address the benefits from retrieval on large scale visual language models.

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

# Appendix

## A  Re-ViLM with Increasing Size of Pretraining and Retrieval Data

In this section, we present detailed results of Re-ViLM in both zero-shot and fine-tuning settings, as shown in Table 6, Table 7, and Table 8. The performance of Re-ViLM continually improves with the increasing size of the pretraining set and retrieval database. As the retrieval database grows, Re-ViLM has a higher chance of obtaining more relevant evidence from the retriever, resulting in better performance.

## B  Re-ViLM with Different Number of Retrieved Captions

In this section, we present detailed results of Re-ViLM with different number of retrieved captions $k$ in both zero-shot and fine-tuning settings on MSCOCO dataset, as shown in Table 9. We can see that, while the number of retrieved examples increases, there is no additional benefit in further improving our results (sometimes even worse). We hypothesis that this is due to i) the original RETRO LM (used to initialize Re-ViLM) is pretrained with $k = 2$, and ii) larger $k$ may return captions that are less related to the test image.

## C  Multiple Images with The Same Caption

In this section, we delve into the issue of data duplication, where multiple images have the identical caption. As illustrated in Figure 3, there are three distinct scenarios: **#1)** different views of the same event, **#2)** identical scenes with different poses, and **#3)** different scenes with the same description. In scenario 1), the provided caption describes an event with a series of pictures taking at the same time, but it is not well related to a specific picture, e.g., the first picture in this scenario. As a result, it is undesired to directly copy the retrieved caption as the model output at both training and test stage. If the retrieved similar image-text pair belongs to scenario **2)** and **3)**, the caption can be useful even with direct copy-and-paste at test time. Specifically, for scenario **2)**, images depict the same scene (e.g. a woman holding a cell phone in an office) and should have highly correlated or even identical caption. However, if the large part of training set consists of such examples [5], the retrieval-augmented models are overly encouraged to copy-and-paste the retrieved caption as the output, which can undermine their generalization ability at test stage, especially for out-of-domain settings.

Table 6: Full zero-shot evaluation results on MSCOCO dataset. **CCS** refers to CC3M+CC12M+SBU dataset and **COYO** the COYO104M. We also conduct experiments on showing "in-domain" retrieval by using **MSCOCO** dataset as retrieval base during inference. This brings further performance improvements. We also present the baseline results by directly evaluating with the retrieved captions from **CCS**+**COYO** database. Results indicate that the retrieved caption itself cannot be directly used for captioning generation.

| Method | Pretraining | | Evaluation | | |
|---|---|---|---|---|---|
| | Training Set | Retrieval Database | Retrieval Database | BLEU@4 | CIDer |
| Retrieved Captions | - | - | CCS+COYO | 0.0 | 3.6 |
| Re-ViLM$_{base}$ | CCS | CCS | CCS | **17.6** | 49.4 |
| Re-ViLM$_{base}$ | CCS+COYO | CCS+COYO | CCS+COYO | 17.0 | 51.2 |
| Re-ViLM$_{base}$ | CCS+COYO | CCS+COYO | MSCOCO | 17.4 | **55.2** |
| Re-ViLM$_{medium}$ | CCS | CCS | CCS | 17.3 | 52.8 |
| Re-ViLM$_{medium}$ | CCS+COYO | CCS+COYO | CCS+COYO | 17.9 | 53.6 |
| Re-ViLM$_{base}$ | CCS+COYO | CCS+COYO | MSCOCO | **18.2** | **57.4** |
| Re-ViLM$_{large}$ | CCS | CCS | CCS | 19.2 | 59.6 |
| Re-ViLM$_{large}$ | CCS+COYO | CCS+COYO | CCS+COYO | 18.6 | 60.8 |
| Re-ViLM$_{base}$ | CCS+COYO | CCS+COYO | MSCOCO | **18.8** | **65.4** |

---

[5]For example, we find the ratio of training examples that have the same captions as others can be as high as 15.1% in Conceptual Captions (Sharma et al., 2018).

Table 7: Full zero-shot evaluation results on Flickr30k dataset. **CCS** refers to CC3M+CC12M+SBU dataset and **COYO** the COYO104M.

| Method | Pretraining | | Evaluation | | |
| --- | --- | --- | --- | --- | --- |
| | Training Set | Retrieval Database | Retrieval Database | CIDer | SPICE |
| Re-ViLM$_{base}$ | **CCS** | **CCS** | **CCS** | 45.0 | **9.2** |
| Re-ViLM$_{base}$ | **CCS**+**COYO** | **CCS**+**COYO** | **CCS**+**COYO** | **45.2** | **9.2** |
| Re-ViLM$_{medium}$ | **CCS** | **CCS** | **CCS** | 50.8 | 9.5 |
| Re-ViLM$_{medium}$ | **CCS**+**COYO** | **CCS**+**COYO** | **CCS**+**COYO** | **52.0** | **9.8** |
| Re-ViLM$_{large}$ | **CCS** | **CCS** | **CCS** | 51.5 | 9.7 |
| Re-ViLM$_{large}$ | **CCS**+**COYO** | **CCS**+**COYO** | **CCS**+**COYO** | **52.1** | **10.0** |

Table 8: Full fine-tuning evaluation results on MSCOCO dataset. **CCS** refers to CC3M+CC12M+SBU dataset and **COYO** the COYO104M.

| Method | Pretraining | | Evaluation | | |
| --- | --- | --- | --- | --- | --- |
| | Training Set | Retrieval Database | Retrieval Database | BLEU@4 | CIDer |
| Re-ViLM$_{base}$ | **CCS** | **CCS** | **CCS** | 37.0 | 127.5 |
| Re-ViLM$_{base}$ | **CCS** | **CCS** | **CCS**+**COYO** | 37.3 | 128.0 |
| Re-ViLM$_{base}$ | **CCS**+**COYO** | **CCS** | **CCS** | 37.2 | 128.0 |
| Re-ViLM$_{base}$ | **CCS**+**COYO** | **CCS** | **CCS**+**COYO** | 37.4 | 128.6 |
| Re-ViLM$_{base}$ | **CCS**+**COYO** | **CCS**+**COYO** | **CCS**+**COYO** | 37.5 | 128.2 |
| Re-ViLM$_{base}$ | **CCS**+**COYO** | **CCS**+**COYO** | **CCS**+**COYO**+COCO | **37.8** | **129.1** |
| Re-ViLM$_{medium}$ | **CCS** | **CCS** | **CCS**+**COYO** | 37.5 | 129.0 |
| Re-ViLM$_{medium}$ | **CCS**+**COYO** | **CCS** | **CCS**+**COYO** | 37.7 | 129.1 |
| Re-ViLM$_{medium}$ | **CCS**+**COYO** | **CCS**+**COYO** | **CCS**+**COYO** | 38.0 | 129.9 |
| Re-ViLM$_{medium}$ | **CCS**+**COYO** | **CCS**+**COYO** | **CCS**+**COYO**+COCO | **38.2** | **131.2** |
| Re-ViLM$_{large}$ | **CCS** | **CCS** | **CCS** | 38.4 | 129.8 |
| Re-ViLM$_{large}$ | **CCS** | **CCS** | **CCS**+**COYO** | 38.1 | 128.4 |
| Re-ViLM$_{large}$ | **CCS**+**COYO** | **CCS**+**COYO** | **CCS**+**COYO**+COCO | **39.4** | **134.2** |

Table 9: Re-ViLM zero-shot evaluation and finetuning evaluation on MSCOCO dataset with different number of retrieved samples $k$. While the number of retrieved examples increases, there is no additional benefit in further improving our results.

| Method | | Zero-shot Evaluation | | Finetuning Evaluation | |
| --- | --- | --- | --- | --- | --- |
| | | BLEU@4 | CIDer | BLEU@4 | CIDer |
| Re-ViLM$_{base}$ | $k=2$ | 17.0 | 51.2 | 37.8 | 129.1 |
| | $k=5$ | 17.0 | 51.4 | 37.5 | 128.4 |
| | $k=10$ | 16.5 | 49.8 | 37.5 | 128.2 |
| Re-ViLM$_{medium}$ | $k=2$ | 17.9 | 53.6 | 38.2 | 131.2 |
| | $k=5$ | 17.6 | 52.2 | 37.8 | 128.6 |
| | $k=10$ | 17.2 | 50.4 | 37.6 | 128.6 |
| Re-ViLM$_{large}$ | $k=2$ | 18.6 | 60.8 | 39.4 | 134.2 |
| | $k=5$ | 18.5 | 57.2 | 38.4 | 129.2 |
| | $k=10$ | 18.2 | 55.8 | 38.2 | 128.8 |

**Scenario #1: Different views from the same event**

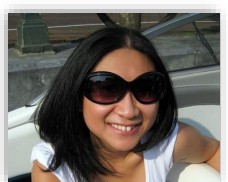 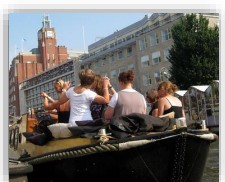 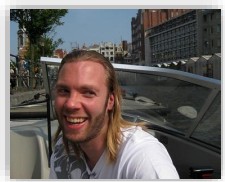

Caption: Hanging out near the flower market bloemenmarkt.

**Scenario #2: Identical scenes with different poses**

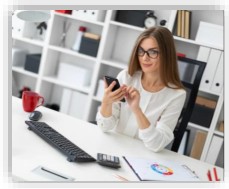 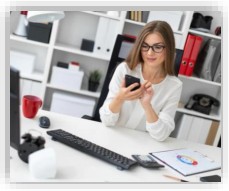 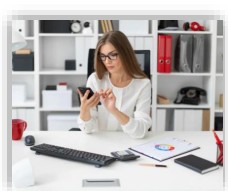

Caption: Women holding a cell phone in the office.

**Scenario #3: Different scenes with the same description**

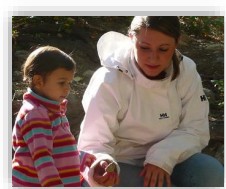 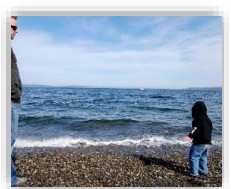 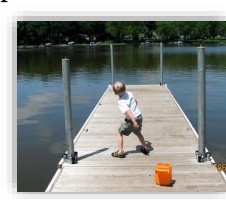

Caption: throwing rocks in the water.

Figure 3: Set of images from conceptual captions dataset with the same captions under **(Top)** Scenario #1): Different views from the same event. **(Middle)** #2): Identical scenes with different poses. **(Bottom)** Scenario #3): Different scenes with the same description. The example of images are from Conceptual Captions (Sharma et al., 2018)