# OpenReview forum: "Re-ViLM: Retrieval-Augmented Visual Language Model for Zero and Few-Shot Image Captioning"
_EMNLP/2023/Conference — EMNLP 2023 Findings_

### Official Review · Reviewer_FVoX · 2023-07-31

**Soundness:** 3

**Excitement:**

3: Ambivalent: It has merits (e.g., it reports state-of-the-art results, the idea is nice), but there are key weaknesses (e.g., it describes incremental work), and it can significantly benefit from another round of revision. However, I won't object to accepting it if my co-reviewers champion it.

**Paper Topic And Main Contributions:**

This paper aims to enhance the 0-shot performance of an image captioning system via retrieval.

**Questions For The Authors:**

- A. The benefit of incorporating retrieval into captioning is not well demonstrated
    - A1. For 0-shot evaluation, the previous method can achieve ~50 CIDEr on MS-COCO by only training on ~1M captions (Table 1). BLIP (ViT-B/16) can achieve 94.8 CIDEr on Flickr30K by training on 14M image-text pairs (Table 1). The authors use more data and larger models, however, no better performance is achieved. So I wonder about the benefit of such a "retrieval augment" technique roadmap.
    - A2. For the finetuning case, Re-ViLM is worse than previous methods, BLIP, and SimVLM.
    - A3. It would be nice if the authors can report the cost time of the retrieval process.  How long will it take to run a kNN algorithm with 700M entries?
    - A4. What is the performance of the method without retrieval?

- B. In Figure 2, we can see the retrieved captions have irrelevant objects with the input images, for example, "woman" and "laptop" are not in the input image. In this case, how does the method guarantee that the output caption will not be affected by these noises?

- C. How can authors guarantee there are not the same samples both in the retrieval database and test set? Is the 0-shot setting really 0-shot?

- D. Let's assume retrieval results are important for the final captioning performance, why only use CLIP-ViT-B/32 to extract the embedings? CLIP-ViT-L is much better than CLIP-VIT-B/32, it would be interesting to see the performance of the method with a CLIP-ViT-L augmented retrieval system.

- E. Reference Lines 603-614 have two identical bib entries.

[1] https://arxiv.org/pdf/2303.03032.pdf

[2] BLIP: Bootstrapping Language-Image Pre-training for Unified Vision-Language Understanding and Generation

**Reasons To Accept:**

- It is interesting to have such a trial of captioning with retrieval.

**Reasons To Reject:**

- The benefit of incorporating retrieval into captioning is not well demonstrated.

**Reproducibility:**

3: Could reproduce the results with some difficulty. The settings of parameters are underspecified or subjectively determined; the training/evaluation data are not widely available.

**Reviewer Confidence:**

4: Quite sure. I tried to check the important points carefully. It's unlikely, though conceivable, that I missed something that should affect my ratings.

---

> ### Author Rebuttal · Authors · 2023-08-29
>
> Thank you so much for your detailed review. We will address your comments in the following.
>
> > "A1, A2. For 0-shot evaluation, the previous method can achieve ~50 CIDEr on MS-COCO by only training on ~1M captions (Table 1). BLIP (ViT-B/16) can achieve 94.8 CIDEr on Flickr30K by training on 14M image-text pairs (Table 1). The authors use more data and larger models, however, no better performance is achieved. So I wonder about the benefit of such a "retrieval augment" technique roadmap. For the finetuning case, Re-ViLM is worse than previous methods, BLIP, and SimVLM."
>
>   - Many thanks for the new reference to the DeCap paper. We didn’t notice this interesting work when we submitted our paper. We will add reference and discuss it in our final version. Regarding your performance concern on Re-ViLM, we should clarify that our paper mainly focuses on proposing a simple and novel retrieval augmentation pipeline, which can be generally applied to many other visual language models. We choose Flamingo as our base model due to its compelling in-context few-shot capability (learned from interleaved text image data) and great potential for scaling (demonstrated by a 80B model in the original paper), although it is not considered as a sample-efficient and parameter-efficient model, especially for the finetuning case. Indeed, the recent OpenFlamingo / MMC4 efforts from the open source community also underscore the importance of studying such architecture and constructing interleaved dataset.
>
> > "A3. It would be nice if the authors can report the cost time of the retrieval process. How long will it take to run a kNN algorithm with 700M entries?"
> - Thanks for pointing this out. With the Faiss library built on a 700M entries database, given a query image, we can retrieve its kNN neighbors in 50ms on average.
>
> > "A4. What is the performance of the method without retrieval?"
> - Sorry for the confusion. The performance of the method, without retrieval, can be shown as follows:
>
> |             MSCOCO Zero-shot                       | BLEU@4 | CIDer |
> |-------------------------------------|:-------------:|:------------:|
> | Re-ViLM (base, with retrieval)      |      17.0     |     51.2     |
> | Re-ViLM (base, without retrieval)   |      9.8      |     36.6     |
> | Re-ViLM (medium, with retrieval)    |      17.9     |     53.6     |
> | Re-ViLM (medium, without retrieval) |      10.6     |     39.2     |
> | Re-ViLM (large, with retrieval)     |      18.6     |     60.8     |
> | Re-ViLM (large, without retrieval)  |      10.2     |     39.6     |
>
> From the results we can see that, Re-ViLM without retrieval could lead to worse performance compared to Re-ViLM with retrieval during inference, even worse compared with Flamingo (re-impl). This is reasonable since we fully utilized retrieved evidence during training and once we cannot obtain any retrieved info during inference, it may cause the train/test mismatch.
>
> > "B. In Figure 2, we can see the retrieved captions have irrelevant objects with the input images, for example, "woman" and "laptop" are not in the input image. In this case, how does the method guarantee that the output caption will not be affected by these noises?"
>
> - Thanks for your question. Re-ViLM utilizes and fuses two sources of information to help generate relevant caption: visual features from visual encoder, and retrieved captions from retriever. While the retrieved information can be informative but noisy, Re-ViLM will unitize the visual feature through the gated cross-attention to discriminate between relevant and irrelevant key word descriptions. Thus the generation will be enriched by retrieved captions (see Figure 1), while still tends to ignore the noises within them.
>
> > "C. How can authors guarantee there are not the same samples both in the retrieval database and test set? Is the 0-shot setting really 0-shot?"
>   - Thanks for bringing up this question. According to [COYO-700M’s official documentation](https://github.com/kakaobrain/coyo-dataset), it states that they “removed all duplicate images based on the image pHash value from external public datasets: ImageNet-1K/21K, Flickr-30K, MS-COCO, CC-3M, CC-12M”, as shown in the Data Filtering section. We have also conducted careful verification showing that there is no leakage between the retrieval database and test set. Based on this, we believe that our 0-shot setting is really 0-shot.
>
> > "D. Let's assume retrieval results are important for the final captioning performance, why only use CLIP-ViT-B/32 to extract the embedings? CLIP-ViT-L is much better than CLIP-VIT-B/32, it would be interesting to see the performance of the method with a CLIP-ViT-L augmented retrieval system."
>   - Thanks for your suggestion. We conduct additional experiments on CLIP-ViT-L retrieval system and results are shown as follows:
>
> |           MSCOCO Zero-shot                    | BLEU@4 | CIDer |
> |---------------------------------|:-------------:|:------------:|
> | Re-ViLM (base, CLIP-ViT-B/32)   |      17.0     |     51.2     |
> | Re-ViLM (base, CLIP-ViT-L/14)   |      16.8     |     50.4     |
> | Re-ViLM (medium, CLIP-ViT-B/32) |      17.9     |     53.6     |
> | Re-ViLM (medium, CLIP-ViT-L/14) |      17.9     |     54.0     |
> | Re-ViLM (large, CLIP-ViT-B/32)  |      18.6     |     60.8     |
> | Re-ViLM (large, CLIP-ViT-L/14)  |      18.4     |     59.2     |
>
>
> From the results, we can see that there is no significant difference between CLIP-ViT-L retrieval system and CLIP-ViT-B system. Considering additional computational and disk storage cost by introducing CLIP-ViT-L, we keep our current ViT-B embedding as retrieval measurement.
>
> > "E. Reference Lines 603-614 have two identical bib entries."
>   - Thanks for pointing this out. We will remove duplicated references in our revision.
>
> We hope our response addresses your concerns.

---

### Official Review · Reviewer_uG9w · 2023-08-03

**Soundness:** 3

**Excitement:**

3: Ambivalent: It has merits (e.g., it reports state-of-the-art results, the idea is nice), but there are key weaknesses (e.g., it describes incremental work), and it can significantly benefit from another round of revision. However, I won't object to accepting it if my co-reviewers champion it.

**Paper Topic And Main Contributions:**

In this paper, the authors introduce a Retrieval-augmented Visual Language Model built upon the Flamingo called "Re-ViLM". Re-ViLM was designed to solve enormous model parameters requirement based on Flamingo. Improvements were made on Re-ViLM:1) storing specific knowledge explicitly in the external database, 2) constructing an interleaved image and text data. Re-ViLM significantly boosts performance for image-to-text generation tasks.

**Reasons To Accept:**

The advantages of this article are as follows:
1. This paper uses existing public datasets to build a pre-training and evaluation dataset consisting of interleaved image-text pairs for multi-modal pre-training, facilitating contextual learning.
2. The Re-ViLM model outperforms baselines on various downstream tasks, and results from ablation studies also support the effectiveness of the proposed improvements.

**Reasons To Reject:**

The weakness of this paper is listed as follows:
1.  The novelty of this paper was limited. The Re-ViLM model basically followed the structure of the state-of-the-art visual LM and a few minor improvements were proposed.
2. The most serious problem with this paper is that there is no section discussing limitations.

**Reproducibility:**

3: Could reproduce the results with some difficulty. The settings of parameters are underspecified or subjectively determined; the training/evaluation data are not widely available.

**Reviewer Confidence:**

4: Quite sure. I tried to check the important points carefully. It's unlikely, though conceivable, that I missed something that should affect my ratings.

---

> ### Author Rebuttal · Authors · 2023-08-29
>
> Many thanks for your comments. We will address your comments in the following.
>
> > “The novelty of this paper was limited. The Re-ViLM model basically followed the structure of the state-of-the-art visual LM and a few minor improvements were proposed.”
>
> Regarding your novelty concern, we address our opinion as follows:
>   - First, the way we introduced our retrieval modules is novel: we design our own retrieval algorithm with filtering to carry the retrieved info. into language model generation through cross-attention modules, which hasn’t been done in multimodal training before.
>   - Second, our data processing is carefully designed. In multi-modal datasets, there are cases where multiple captions describing the same image and multiple images having the same caption. Thus, simply performing standard nearest neighbor retrieval tends to make the model take a shortcut and copy-paste retrieval examples during training, hurting the model performance.
>   - Third, the interleaved dataset we constructed to empower the in-context few-shot learning capability of the model is novel and important to the community.  The original Flamingo paper crawls and constructs an in-house interleaved <image, text> dataset. However, it is not publicly available and really expensive to collect such a dataset. In this work, we propose a novel method to construct such an interleaved dataset from publicly available paired <image, text> dataset (see section 4.2).
>
>
> > “The most serious problem with this paper is that there is no section discussing limitations.”
>
> We sincerely thank you for this reminder. We will include the limitations section in our paper as follows:
>   - In this paper, we focus on exploring emergent zero-shot and in-context few-shot image captioning. To achieve this, we designed our retrieval augmented model mainly based on the Flamingo framework (Alayrac et al., 2022), and leave the application of our retrieval design to other image-to-text frameworks (e.g., Bao et al., 2021; Chen et al., 2022c; Wang et al., 2022a) as future work. Furthermore, since there is no official implementation of Flamingo and its training datasets, our framework is based on our reimplemented Flamingo, trained on publicly available datasets and manually crafted interleaved image-text datasets. Also from the scaling perspective, comparable to GPT in the text-only domain, one of the most important advantages of the Flamingo-like model is scaling. In this study, we haven’t been able to further scale Re-ViLM to 80B to address the benefits from retrieval on large scale visual language models.
>
> [1] Alayrac, et al. "Flamingo: a visual language model for few-shot learning." NeurIPS 2022.
>
> [2] Bao, et al. "Beit: Bert pre-training of image transformers." arXiv:2106.08254, 2021.
>
> [3] Chen, et al. "Pali: A jointly-scaled multilingual language-image model." arXiv:2209.06794, 2022c.
>
> [4] Wang, et al. "Git: A generative image-totext transformer for vision and language." arXiv:2205.14100, 2022a.
>
>
> We hope our response addresses your concerns.

---

### Official Review · Reviewer_ag8H · 2023-08-05

**Soundness:** 3

**Excitement:**

3: Ambivalent: It has merits (e.g., it reports state-of-the-art results, the idea is nice), but there are key weaknesses (e.g., it describes incremental work), and it can significantly benefit from another round of revision. However, I won't object to accepting it if my co-reviewers champion it.

**Paper Topic And Main Contributions:**

The model proposes a new method which utilizes retrieved knowledge for image captioning task.

**Questions For The Authors:**

Why does Re-ViLM base obtain BLEU@4 of 14.8 in the 2-shot setting?

**Reasons To Accept:**

A new method is proposed, by building the model based on pre-trained models and utilizing retrieval-augmented knowledge, the proposed method leads to better results across different settings including zero-shot, few-shot and fully fine-tuning. Some ablation studies are also conducted.

**Reasons To Reject:**

One important baseline is needed, which is directly evaluating with the retrieved captions (with the proposed filtering). Meanwhile, more examples of retrieved examples with corresponding generated results are suggested to be shown.


Results with different number of retrieved images are suggested, which can better illustrate the effectiveness of the proposed architecture and retrieval-augmentation respectively.

Inference time and memory comparisons are not shown, they are suggested because extra time and memory are needed to perform the retrieval.

**Reproducibility:**

4: Could mostly reproduce the results, but there may be some variation because of sample variance or minor variations in their interpretation of the protocol or method.

**Reviewer Confidence:**

3: Pretty sure, but there's a chance I missed something. Although I have a good feel for this area in general, I did not carefully check the paper's details, e.g., the math, experimental design, or novelty.

---

> ### Author Rebuttal · Authors · 2023-08-29
>
> Thank you so much for your comments and valuable suggestions. They are really helpful for improving our paper.  We will address your comments in the following.
>
> > “One important baseline is needed, which is directly evaluating with the retrieved captions (with the proposed filtering). ”
>   - Many thanks for pointing it out. We conduct the suggested evaluation on MSCOCO and show the results here.  We perform retrieval on the COYO+CCS dataset by choosing the most similar captions for each testing image under ViT-B/32 embedding space after filtering. The BLEU@4 and CIDer score is 0.0 and 3.6 respectively, which means the retrieved caption itself cannot be directly used for captioning generation. We will add this result as an important baseline into our revision.
>
> > “More examples of retrieved examples with corresponding generated results are suggested to be shown.”
>   - Many thanks for your nice suggestion. We think it’s very important. We will include more examples in the final version.
>
> > “Results with different number of retrieved images are suggested”
>   - Many thanks for this suggestion. We follow your suggestion and conduct the experiment.  We evaluated on $k={5, 10}$ settings on each size of the Re-ViLM model. While the number of retrieved examples increases, there is no additional benefit in further improving our results (sometimes even worse). We hypothesis that this is due to i) the original RETRO LM (used to initialize Re-ViLM) is pretrained with  $k=2$, and ii) larger $k$ may return captions that are less related to the test image. We will add these results into the ablation study section in our revision.
>
> |        MSCOCO Zero-shot                | BLEU@4 | CIDer |
> |------------------------|:-------------:|:------------:|
> | Re-ViLM (base, k=2)    |      17.0     |     51.2     |
> | Re-ViLM (base, k=5)    |      17.0     |     51.4     |
> | Re-ViLM (base, k=10)   |      16.5     |     49.8     |
> | Re-ViLM (medium, k=2)  |      17.9     |     53.6     |
> | Re-ViLM (medium, k=5)  |      17.6     |     52.2     |
> | Re-ViLM (medium, k=10) |      17.2     |     50.4     |
> | Re-ViLM (large, k=2)   |      18.6     |     60.8     |
> | Re-ViLM (large, k=5)   |      18.5     |     57.2     |
> | Re-ViLM (large, k=10)  |      18.2     |     55.8     |
>
> |     MSCOCO Finetuning                  | BLEU@4 | CIDer |
> |------------------------|:-------------:|:------------:|
> | Re-ViLM (base, k=2)    |      37.8     |     129.1    |
> | Re-ViLM (base, k=5)    |      37.5     |     128.4    |
> | Re-ViLM (base, k=10)   |      37.5     |     128.2    |
> | Re-ViLM (medium, k=2)  |      38.2     |     131.2    |
> | Re-ViLM (medium, k=5)  |      37.8     |     128.6    |
> | Re-ViLM (medium, k=10) |      37.6     |     128.6    |
> | Re-ViLM (large, k=2)   |      39.4     |     134.2    |
> | Re-ViLM (large, k=5)   |      38.4     |     129.2    |
> | Re-ViLM (large, k=10)  |      38.2     |     128.8    |
>
>
> > “Inference time and memory comparisons are not shown, they are suggested because extra time and memory are needed to perform the retrieval.”
>   - This is a nice suggestion. At inference, we utilize the Faiss index for fast similarity search, which takes ~50 ms per query for a database with 700M text-image pairs. The Faiss index is 241 GB. Note that the retrieval is performed on the CPU and the whole index is only loaded into CPU memory.
>
> > “Why does Re-ViLM base obtain BLEU@4 of 14.8 in the 2-shot setting?”
>   - Thanks for raising the question. Re-ViLM obtains BLEU@4 of 17.0 and CIDer of 51.2 in zero-shot setting, while worse BLEU@4 of 14.8 and better CIDer of 60.1 in 2-shot setting. We do notice this discrepancy between BLEU@4 and CIDer sometimes, especially when both of them are relatively low. This is not surprising, as BLEU@4 focuses on 4-gram, while CIDer averages {1, 2, 3, 4}-gram similarities. We will elaborate this in our revision.
>
> We really appreciate your valuable suggestions. They are very helpful for improving our paper. We follow your suggestions and hope our response addresses your concerns.

---

### Meta-Review · Area_Chair_2YfU · 2023-09-11

**Recommendation:** 3

**Metareview:**

The paper proposed the VL models with augmentation ability, trying to use <image, text> paired data as an additional database to augment the model's knowledge. The experiments are comprehensive with multiple experimental setups. The method is new to the reviewer while there is a discussion regarding the novelty of the method. Also, the reviewer also pointed out that the effectiveness of this method is not fully demonstrated with the empirical results.

---

### Decision · Program_Chairs · 2023-10-07

**Decision:**

Accept-Findings

**Comment:**

The paper proposed the VL models with augmentation ability, trying to use <image, text> paired data as an additional database to augment the model's knowledge. The experiments are comprehensive with multiple experimental setups. The method is new to the reviewer while there is a discussion regarding the novelty of the method. Also, the reviewer also pointed out that the effectiveness of this method is not fully demonstrated with the empirical results.